# Ursodeoxycholic Acid Binds PERK and Ameliorates Neurite Atrophy in a Cellular Model of GM2 Gangliosidosis

**DOI:** 10.3390/ijms24087209

**Published:** 2023-04-13

**Authors:** Carolina Morales, Macarena Fernandez, Rodrigo Ferrer, Daniel Raimunda, Dolores C. Carrer, Mariana Bollo

**Affiliations:** Instituto de Investigación Médica M y M Ferreyra, INIMEC-CONICET, Universidad Nacional de Córdoba, Córdoba 5016, Argentina

**Keywords:** lysosomal storage disease, chemical chaperones, ATP binding pocket

## Abstract

The Unfolded protein response (UPR), triggered by stress in the endoplasmic reticulum (ER), is a key driver of neurodegenerative diseases. GM2 gangliosidosis, which includes Tay-Sachs and Sandhoff disease, is caused by an accumulation of GM2, mainly in the brain, that leads to progressive neurodegeneration. Previously, we demonstrated in a cellular model of GM2 gangliosidosis that PERK, a UPR sensor, contributes to neuronal death. There is currently no approved treatment for these disorders. Chemical chaperones, such as ursodeoxycholic acid (UDCA), have been found to alleviate ER stress in cell and animal models. UDCA’s ability to move across the blood-brain barrier makes it interesting as a therapeutic tool. Here, we found that UDCA significantly diminished the neurite atrophy induced by GM2 accumulation in primary neuron cultures. It also decreased the up-regulation of pro-apoptotic CHOP, a downstream PERK-signaling component. To explore its potential mechanisms of action, in vitro kinase assays and crosslinking experiments were performed with different variants of recombinant protein PERK, either in solution or in reconstituted liposomes. The results suggest a direct interaction between UDCA and the cytosolic domain of PERK, which promotes kinase phosphorylation and dimerization.

## 1. Introduction

The symptoms of neurodegenerative diseases are caused by selective alterations in the functionality of diverse neuronal populations. In the past decade, several studies have shown that endoplasmic reticulum (ER) stress performs a crucial role in many of these pathologies [1,2]. ER stress occurs when perturbations that compromise protein-folding capacity in the organelle trigger a complex signaling, known as the unfolded protein response (UPR). In mammalian cells, UPR consists mainly of three branches initiated by three ER transmembrane sensors: inositol-requiring enzyme 1 (IRE1), activating transcription factor 6 (ATF6), and protein kinase RNA [PKR]-like ER kinase (PERK). These stress sensors activate a fast-signaling response that reduces the influx of proteins into the organelle and reestablish homeostasis. If ER stress remains irremediably high, alternative ‘‘terminal UPR’’ signals induce apoptosis. Chronic or unresolved UPR has been linked with neurodegeneration [3]. PERK leads the only UPR branch that remains active under prolonged ER stress; thus, it mediates both pro-survival and apoptotic responses [4,5,6].

During the acute phase, activated PERK inhibits overall protein synthesis by phosphorylating the α subunit of the eukaryotic translation initiation factor 2α (eIF2α), which reduces the load of proteins at the ER [7]. In addition, PERK phosphorylates NRF2 (nuclear factor-E2 related factor 2), the master regulator of the oxidative stress response that activates a pro-survival signal [8,9]. PERK, a type I ER membrane protein, has serine/threonine kinase activity in its cytosolic domain, which is connected by a transmembrane segment with a stress-sensing luminal domain [7]. Under normal conditions, the luminal portion associates with the ER chaperone BiP, which maintains PERK in its monomeric form and prevents the activation of its downstream signaling. During ER stress, the unfolded protein accumulated in the lumen competitively titrates BiP from PERK, which causes the latter to oligomerize and become active [10,11]. According to emerging evidence, the activation of UPR also involves alterations in the lipid composition of the ER membrane, independently of any sensing by the luminal domain [12,13].

Several cytosolic proteins physically associate with UPR sensors and modulate the amplitude of their downstream signaling. P58^IPK^, a member of the Hsp40 family, directly interacts with PERK and inhibits its kinase activity [14]. Earlier research conducted by our group demonstrated that calcineurin, a Ca^2+^ dependent phosphatase, also associates with the cytosolic domain of PERK, and promotes its autophosphorylation and downstream signaling [15,16].

GM2 gangliosidosis is a neurodegenerative storage disorder that includes Sandhoff and Tay-Sachs disease. It is characterized by a deficiency in β-N-acetylhexosaminidase (Hex) activity, which impairs the degradation of GM2, a complex lipid ganglioside. The time of onset and the severity of the clinical manifestations are correlated with the degree of residual Hex activity. Nevertheless, patients are clinically normal at birth and the symptoms always consist of progressive neurodegeneration. In the infantile form of the disease, clinical signs appear at ∼3–5 months of age, and they include motor deficit, visual deterioration, and developmental arrest, among others. Death usually occurs at age 2 [17,18]. Similar symptoms (mental and motor retardation) are typical in the late-infantile and juvenile forms. They worsen over time and life expectancy does not surpass 40. There is considerable pathological and biochemical evidence that cerebral degeneration is marked by the progressive accumulation of GM2.

The molecular mechanisms through which GM2 buildup leads to neuronal death are still not fully understood. However, we recently found that PERK activation partly mediated neuronal death in a cellular model of GM2 gangliosidosis [6]. Excessive accumulation of this ganglioside in the ER membrane decreased the luminal ER Ca^2+^ content. In turn, this activates PERK signaling, which shows two phases, one acute/cytoprotective and the other apoptotic, during which the expression of the pro-apoptotic transcription factor C/EBP homologous protein (CHOP) is enhanced. BiP overexpression partially reverted GM2-dependent ER Ca^2+^ depletion and consequently PERK phosphorylation. This means that the conventional stress-sensing mechanism was active in this GM2-gangliosidosis cell model, but alterations in the lipid composition of the ER may have contributed to PERK activation. Irrespective of its mechanism of activation, it is clear that PERK is a key factor to change neuron vulnerability to apoptosis associated with GM2 accumulation.

Despite considerable efforts, there is currently no approved therapy for neuronal ganglioside storage diseases [19]. In recent years, strategies to manipulate UPR have become attractive in the design of novel treatments of neurodegenerative diseases. One way to do so would be through orally active chemical chaperones. These are a group of structurally unrelated compounds with low molecular weight, which stabilize protein conformation and improve ER folding capacity and vesicular trafficking [20,21,22]. Ursodeoxycholic acid (UDCA), a bile acid, is part of this group of molecules, and has been reported to modulate ER functions in in vitro and in vivo models [23,24,25,26,27]. This has been approved by the US Food and Drug Administration (FDA) for the treatment of various disorders.

In the present study, we found that UDCA decreases the up-regulation of pro-apoptotic CHOP and neurite atrophy induced by the abnormal accumulation of GM2 in primary neuron cultures. Moreover, when exploring alternative UDCA mechanisms of action, we discovered that it might directly interact with the cytosolic domain of PERK, which promotes kinase auto-phosphorylation and stabilizes its dimerization.

## 2. Results

### 2.1. UDCA Treatment Reverses the Decreased Neurite Outgrowth following GM2 Loading

Increasing evidence of a strong link between ER stress, in particular PERK signaling, and neurodegenerative diseases suggests the possibility of manipulating UPR for therapeutic applications. The present study explored the putative action of UDCA. First, we immunostained primary rat neuronal cultures with microtubule-associated protein 2 (MAP-2) [28], to find out whether this endogenous bile acid had a protective effect against neuritic atrophy caused by GM2 accumulation. Consistent with our previous report, after 20 h of incubation with GM2 there was a significant decrease in total neurite outgrowth attached to the cell bodies with respect to total cell number. This reduction was significantly reverted after treatment with 200 µM and 50 µM of UDCA (Figure 1). A similar cytoprotective effect was observed when cells, in which neuritic atrophy was later induced, were pre-treated with the steroid (Appendix A). These data show that UDCA protects against experimental GM2-gangliosidosis in cultured cells.

### 2.2. UDCA Fits into the ATP-Binding Site of mPERK KD

Anti-aggregation properties have been ascribed to UDCA and other compounds with low molecular weight and an amphiphilic structure. The relief observed in ER stress when cellular and animal models were treated with these substances might have been due to a reduction in protein misfolding and in the accumulation of aggregates, the most upstream event in the UPR [21,22].

However, the fact that lower concentrations were just as effective here and in other studies using cellular and in vivo models [23,29,30,31] suggests a more complex mode of action, probably at different levels in the UPR pathway. In this respect, several small molecules, such as flavonoids and steroids, are known to modulate the activity of various kinases by engaging their nucleotide-biding site (ATP pocket) [32,33]. To examine whether UDCA might do the same with PERK, we manually modeled a UDCA molecule into the structure of active mPERK KD. This was completed on the basis of available models of the mPERK-AMP-PNP complex structure [33,34] and computational predictions (the easy interface mode on HADDOCK 2.4). The ATP-biding site in mPERK KD was clearly able to accommodate the molecule (Figure 2), which means that UDCA could, in principle, directly interact with the cytosolic domain and modulate PERK oligomerization and subsequent activation.

### 2.3. UDCA Prompts PERK Activation at Cellular Level and In Vitro

We assessed the ability of UDCA to promote the activation of PERK and stabilize its active form in primary cortical neuron cultures. The steroid bile acid concentrations chosen were those that reduced neurite atrophy following GM2 accumulation. Immunofluorescence and confocal microscopy showed that when the cells were incubated with UDCA only, there was a significant increase in the levels of phosphorylated PERK (P-PERK) (Figure 3).

To further test our hypothesis that UDCA is a PERK modulator and that a luminal-independent signal is involved in the process, we used liposomes containing the recombinant protein PERK without the luminal domain that senses unfolded protein (ΔLD-PERK). For this, the transmembrane and cytosolic domains of PERK was expressed in *E. coli* as a hexahistidine fusion protein (6xHis-ΔLD-PERK), purified, and integrated into liposomes whose lipid composition either mimicked that of the normal ER membrane (control), or which contained 0.1 mole% of GM2, as was observed previously in our cellular model [6]. The orientation of ΔLD-PERK into liposomes was assessed by determining how susceptible it was to digestion by protease K. The protein was accessible to the protease in all cases, which indicates that its cytosolic domain faced the external liposomal medium (Figure 4, line 3). 

An in vitro kinase assay was conducted to evaluate the autophosphorylation of PERK. The proteoliposomes were incubated with ATP-Mg^2+^ for 30 min, ran on SDS-PAGE, and analyzed by Western blot with a homemade anti-PERK^UT^ antibody [6] that recognizes the higher mobile non-phosphorylated PERK and its phosphorylated form. This assay revealed, first, that the level of PERK phosphorylation was higher in vesicles containing GM2 than in controls (Figure 4b), suggesting that the transmembrane and juxtamembrane domains detected changes in lipid composition. Second, this kinase domain displayed a significant increase in its phosphorylation by the addition of UDCA, used at a concentration that has a neuroprotective effect (Figure 1, Figure 2 and Appendix A). However, this experimental design did not enable us to discriminate whether this effect is due to a direct interaction with the cytosolic domain of PERK, and/or UDCA actually induces changes in lipid vesicle properties, promoting PERK dimerization.

To elucidate this, we performed another in vitro kinase assay with a much simpler model consisting of a GST fusioned to the cytosolic domain of PERK. GST-cytPERK was incubated with ATP-Mg^2+^ and increasing concentrations of UDCA (Figure 4f). The proteins were resolved on SDS-PAGE and immunodetected using the same anti-PERK antibody as in the previous assay (Figure 4a). In solution, the recombinant protein GST-cytPERK exhibits non-phosphorylated and phosphorylated forms, due to trans-autophosphorylation occurring inside the bacteria used to express the construct. However, after the addition of UDCA to the protein in solution (in the absence of lipids), the band intensity corresponding to the phosphorylated form significantly increased, suggesting a direct interaction with the kinase.

### 2.4. UDCA Modulates PERK Signaling Activated by GM2 Accumulation

We examined CHOP levels to determine whether UDCA induced a pro-survival effect mediated by PERK under our experimental conditions. As we have previously shown, at 20 h after GM2 incubation the transcription factor is significantly upregulated and translocated to the nuclei in both Neuro 2a (N2a) neuroblastoma cells (Appendix A) and primary cortical neuron cultures, relative to unstressed cells (Figure 5). Interestingly, the rise in CHOP and its translocation to the nuclei were significantly slowed down after treatment with 200 µM and 50 µM of UDCA (Appendix A and Figure 5). Thus, UDCA’s efficiency at delaying the upregulation of CHOP levels could explain, at least in part, its protective effect against neurite atrophy in GM2-stressed cells.

## 3. Discussion

The findings presented here show that the steroid UDCA protects neurons against stress caused by the accumulation of GM2. Earlier studies proposed that, irrespective of the concentration used, UDCA and other low molecular weight compounds relieve heightened stress in the ER by stabilizing its protein conformation, and thus, improving its folding capacity. Instead of that unspecific anti-aggregation effect, we observed that these benefits are due to UDCA directly and specifically acting on PERK modulation.

Previously, we demonstrated that GM2 activates PERK signaling, which mediates both a pro-survival and an apoptotic response [6]. It is well known that PERK activation is preceded by its dimerization/oligomerization, which promotes the autophosphorylation of its C-terminal cytoplasmic domain at several residues, including Thr980 on the kinase activation loop [34]. As observed in the related eIF2α kinase, PKR, phosphorylation stabilizes the back-to-back dimer and consequently both the activation loop and the α–G helix become ordered. In the case of the latter, this arrangement allows it to bind to eIF2α [7,35,36]. Interestingly, the ATP-biding pocket is empty in the active PERK (dimer) [36]. This pocket can accommodate a variety of hydrophobic ligands, such as sterols and flavonoids [37]. According to our in silico analysis, UDCA can dock perfectly into the nucleotide-binding cleft of PERK. Conformational perturbations caused by ligands in the ATP pocket have been reported to modulate kinase activity, either negatively or positively [32,33,38]. We found that UDCA acts as a positive effector, increasing PERK phosphorylation in cells and in vitro experiments, using two variants of PERK recombinant proteins. Thus, in bulk solution, the cytosolic domain was oligomerized and significantly more phosphorylated after interacting with UDCA. This effect was more evident in a PERK mutant lacking its luminal domain (ΔLD-PERK), which had been incorporated into liposomes, which allows productive dimerization or higher-order oligomerization and more potent auto-phosphorylation of the PERK cytosolic domain. Although we cannot exclude the possibility that UDCA may also affect lipid vesicle fluidity, prompting PERK dimerization indirectly, the improvement of a dimer self-association on the membrane surface has been reported for the transmembrane epidermal growth factor receptor (EGFR) [39].

Furthermore, low micromolar concentrations of UDCA were effective at associating directly with PERK and inducing its activation, both in vitro and in cellular-based approaches. In contrast, other studies attribute a protein anti-aggregation effect to UDCA and other similar compounds. In this case, protein folding capacity is enhanced, ER stress is alleviated, and there is a decrease (instead of an increase) in the activation of UPR sensors [23,29,30,31]. In these studies, millimolar concentrations of the compounds were required to achieve the anti-aggregation effect, which involves the interaction of molecules with amphiphilic structures, such as UDCA, with exposed hydrophobic segments of the unfolded proteins [21,40,41].

We also noted that UDCA markedly reduced the pro-apoptotic translocation of CHOP into the nuclei induced by GM2 accumulation. In other words, it appears that within the cellular context created for the experiment, treatment with the bile acid pushed GM2-induced PERK signaling towards cell survival. Similarly, we had observed before that PERK signaling regulation can make cells less vulnerable to neurite atrophy induced by GM2 buildup [6]. The delay in CHOP upregulation might have responded to an activation of the anti-oxidant NRF2 pathway. The same mechanism was identified for compound A4, another PERK activator, in mouse embryo fibroblast (MEF) cells stressed with thapsigargin [42].

It is known that GM2 is not a normal component of the ER membrane [43]; however, this complex sphingolipid appears as a constituent in cellular [6] and in animal [44] GM2-gangliosidosis models. Previously, we found that GM2 accumulation induces a BiP-mediated classical UPR, and thus, ER calcium depletion activates PERK in rat cultured neurons [6]. Here, we designed an experimental system in which ΔLD-PERK was incorporated into liposomes whose composition mimicked that of ER membranes loaded with GM2 [6,45,46], we observed that lipid perturbation also enhances PERK dimerization, independently of the luminal domain. We have used only 0.1 mole% of ganglioside and the lipid mixture should be in the liquid disordered phase at the temperature of the in vitro kinase assays (27 °C) [47]. In this condition, it is unlikely that any effect of the ganglioside on PERK oligomerization is produced by changes described before and that are brought about by an increase in membrane order by saturated acyl chains [13]. However, the formation of nanodomains of GM2 in the membrane cannot be entirely ruled out, even at the low concentrations used [48,49]. The polar head groups in gangliosides can protrude 1–1.5 nm into the aqueous phase, beyond the phospholipids head groups [50]. In our reconstituted liposomes, the ganglioside polar head group (more precisely, the carboxyl group of its sialic acid residue) might have interacted electrostatically with the basic amino acid residues from the juxta membrane domain of PERK [7], and enrich the first lipid shell surrounding the protein with GM2. In turn, this may have induced conformational changes in PERK that promoted its dimerization. Given the selectivity of the interactions between lipid headgroups and proteins, a change in the former could significantly shift the conformational equilibrium of the latter [51]. 

These results, together with previous ones reported by our group [6] and others [10,52], suggest that dimerization of full-length PERK is initiated by the stress-sensing luminal domain, and that perturbations in lipid membrane composition could further stimulate kinase activation. In addition, UDCA can directly interact with the cytosolic domain of PERK, which engages the ATP pocket, stabilizing the active form of the kinase. This mechanistic link may be exploited for therapeutic purposes, since orally active UDCA clearly reduced neurite atrophy in our cellular model of GM2 gangliosidosis.

## 4. Materials and Methods

### 4.1. Reagents and Antibodies

Dulbecco’s Modified Eagle’s Medium (DMEM) (#11995065), Neurobasal Medium (#21103049), trypsin-EDTA 0.25% (#25200056), penicillin-streptomycin (#15140122), B27 Supplement (#17504944), and Glutamax (#35050061) were from Life Technologies (Carlsbad, CA, USA), Fetal calf serum (FCS) was from Internegocios SA (Mercedes, Argentina). The protease inhibitors, purchased from Santa Cruz Biotechnology (Dallas, TX, USA), were leupeptin hemisulfate (#295358), pepstatin A (#45036), aprotinin (#3595), and phenylmethylsulfonyl fluoride (#329-98-6). Monosialoganglioside GM2 (#1502) was obtained from Matreya (State College, PA, USA). The primary antibodies used were anti-P-PERK (#3179), anti-CHOP (#L63F7) (Cell Signaling Technology, Danvers, MA, USA), anti-CHOP (#MA1-250) (Thermo Fisher, Waltham, MA, USA), anti-MAP-2 (#PCK-554P) (Covance Inc. Princeton, NJ, USA), and anti-MAP-2 (#M2320) (Sigma, St. Louis, MO, USA). 

### 4.2. Cell Cultures

The cell culture of rat cortical neurons was performed as previously described [53]. Briefly, the cortices of E18 CD1 rat embryos were dissected and dissociated into single cells through enzymatic digestion for 15 min at 37 °C with 0.25% trypsin-EDTA. Digestion was stopped by adding DMEM with 10% FCS. The pellet was washed three times with Ca^2+^/Mg^2+^-free Hanks’ Buffered Salt Solution and 5.5 mM glucose and triturated by passing the cell suspension up and down a 1000-μL pipette tip ten times. Cell suspension was plated at a density of 500 cells/mm^2^ on culture dishes coated with poly-L-lysine. The culture medium was DMEM, supplemented with 8.2 mM glucose, 1% Glutamax, 10% FCS, 1% penicillin, and 1% streptomycin. After 2 h, this medium was replaced with Neurobasal Medium containing B27 Supplement. The cultures were maintained in vitro for 10 days (10 DIV) at 37 °C, in a humidified atmosphere with 5% CO_2_. 

The Neuro2a (N2a) mouse neuroblastoma cells (CCL-131; ATCC, Manassas, VA, USA) were cultured as described previously [6]. Briefly, the cells were grown in DMEM supplemented with 10% FCS, 1% Glutamax, 1% penicillin, and 1% streptomycin at 37°C in a humidified atmosphere with 5% CO_2_. Differentiation was achieved by substituting this medium for DMEM supplemented with 1% FCS and 9% Opti-MEM. 

### 4.3. Immunocytochemistry

The immunofluorescent detection of MAP-2, P-PERK, and CHOP was performed as previously described [6]. Co-localization and P-PERK detection images were obtained with a 60× oil lens (NA 1.4) in a Zeiss confocal microscope LSM 800. For the MAP-2 detection images, a 20× lens of the same microscope was used.

### 4.4. Bacterial Protein Expression

To transform *E. coli* strain BL21, we used bacterial expression plasmid pET-28a-ΔLD-PERK. This plasmid encodes a C-terminal, 6-histidine-tagged fragment of mouse PERK that lacks the luminal domain but contains the cytosolic kinase and transmembrane domains (amino acids 503-1114). The expression and purification were performed essentially as described by Volmer et al. [13]. In brief, following inoculation 280 mM glucose and 1 mM isopropylthio-β-galactoside (IPTG) were added. The bacteria were cultured for 2 h at 37 °C, then for 6 h more at 20 °C. They were resuspended in homogenization buffer (25 mM Tris-HCl pH 7.5, 100 mM sucrose, 300 mM NaCl, 20mM imidazole, 18 mM Triton-x 100, and the protein inhibitors listed in 4.1), and processed 4 to 6 times in a microfluidizer (Avestin Emulsiflex C-3). The lysate was centrifuged at 100,000× *g* for 1 h at 4 °C. The supernatant was incubated with Ni-NTA resin for 2 h at 4 °C and washed with at least 10 volumes of wash buffer (50 mM Tris-HCl pH 7.5, 300 mM NaCl, 30 mM imidazole, 1.8 mM Triton-x100, and the protein inhibitors listed in 4.1). Then, the protein was eluted in modified wash buffer with 300 mM imidazole and no protein inhibitors. All the elutions were dialyzed in buffer with 100 mM Tris-HCl pH 7.5, 2 mM MgCl_2_, 70 mM KCl, 3 mM 2-mercaptoethanol, and 1.8 mM Triton-x100, then stored at −80 °C until they were used.

GST-cytPERK was purified as described previously [15].

### 4.5. Preparation of Liposomes with Incorporated ΔLD-PERK

Unilamellar liposomes were obtained by mixing 1-palmitoyl-2-oleoyl-*sn*-glycero-3-phosphocholine (POPC), Dioleoyl phosphoethanolamine (DOPE), 1,2-dioleoyl-*sn*-glycero-3-phosphoserine (DOPS), and cholesterol at a 65:20:10:5 mole ratio. The same mixture was used to prepare liposomes containing 0.1 mole% of GM2 (with GM2 replacing POPC molecules in the mixture). The lipids were mixed in chloroform:methanol 2:1 v:v. The solvent was evaporated under a current of nitrogen and the lipid film was subjected to high vacuum for 2–4 h. The lipids were then hydrated in buffer (40 mM Tris-HCl pH 7.5, 100 mM NaCl, 2 mM MgCl_2_) at room temperature, vortexed, and extruded 21 times through 0.1 µm polycarbonate membranes with a mini-extruder (Avanti Polar Lipids). 

The liposomes were destabilized by adding Triton X-100 (0.6% *w/v*) and mixed with ΔLD-PERK at a lipid: protein ratio of 80:1 (*w*/*w*). This mixture was incubated for 5 min at 20 °C, then for 15 min at the same temperature with 0.25 mL of pre-washed Bio-beads SM-2 (Bio-Rad) while shaking. The Bio-beads were filtered and replaced with new ones every 15 min. This was completed three times. A fourth incubation with new Bio-beads was carried out overnight at 4 °C. The proteoliposomes were collected after separating the Bio-beads by centrifugation (100,000× *g* for 1 h at 4 °C), and they were used immediately in the kinase assays.

### 4.6. In Vitro Kinase Assays

Autophosphorylation and dimerization/oligomeryzation of ΔLD-PERK was autophosphorylated and GST-cytPERK, respectively, were performed as follows: ΔLD-PERK proteoliposomes containing and not containing GM2 were incubated for 20 min at 27 °C in a 25.2 µL reaction mixture (40 mM Tris HCl pH 7.5, 2 mM MgCl_2_, and 1 µM ATP) with or without 200 µM UDCA. Reactions were stopped by adding Laemmli Buffer and boiling, and proteins were resolved in a 10% SDS-PAGE. GST-cytPERK with and without UDCA were incubated for 5 min at 27 °C in a 25 µL reaction mixture (40 mM Tris HCl pH 7.5, 10 mM MgCl_2_, 100 mM NaCl, 5% glycerol, 1 mM DTT) and 0.1 mM ATP. The reaction was stopped with Laemmli buffer, boiled, and the proteins resolved on 10% SDS-PAGE. 

### 4.7. GST-cytPERK Crosslinking Assay

The GST-cytPERK was incubated in a buffer (80.2 mM Na_2_HPO_4_, 19.8 mM NaH_2_PO_4_, pH 7.4, 2 mM MgCl_2_, 70 mM KCl, and 1 mM DTT) for 20 min at 27 °C with different UDCA concentrations (50 µM, 200 µM, and 1000 µM, respectively), then for 30 min at 27 °C in the presence of 0.4 mM of crosslinker DSS. The reactions were quenched with 1M Tris-HCl pH 7.5 for 15 min at 27 °C. The protein complexes were analyzed on 7% SDS-PAGE and detected with a homemade anti-PERK antibody. 

### 4.8. Western Blotting

Western Blotting was performed as described previously [6]. 

### 4.9. Protein-Ligand Docking

Molecular docking was performed using HADDOCK 2.4 in the Easy interface mode (https://wenmr.science.uu.nl/haddock2.4/ (accessed on 15 February 2022)) [54]. The starting structural atomic coordinates belonged to mPERK kinase domain (pdb entry 3QD2) [34] and the ursodeoxycholic acid (UDCA) was extracted from bile acid-binding protein (AKR1C2) [55] (pdb entry 1IHI). Active residues in mPERK KD were selected manually in USCF Chimera [56] after obtaining a modeled mPERK-AMP-PNP, created by superposition of the mPERK KD structure and PRK-AMP-PNP (2A19) [33]. HADDOCK parameters were left unchanged, except for the selection of active residues. Finally, the four models with highest scores were selected to depict the putative binding pocket of UDCA in mPERK KD using UCSF Chimera. 

### 4.10. Data Analysis

The results are presented as the means of independent replicates of the experiments ± SEM. The data were statistically analyzed on GraphPad Prism 8.0. One-way analysis of variance (ANOVA) and Tukey test were performed. 

## Figures and Tables

**Figure 1 ijms-24-07209-f001:**
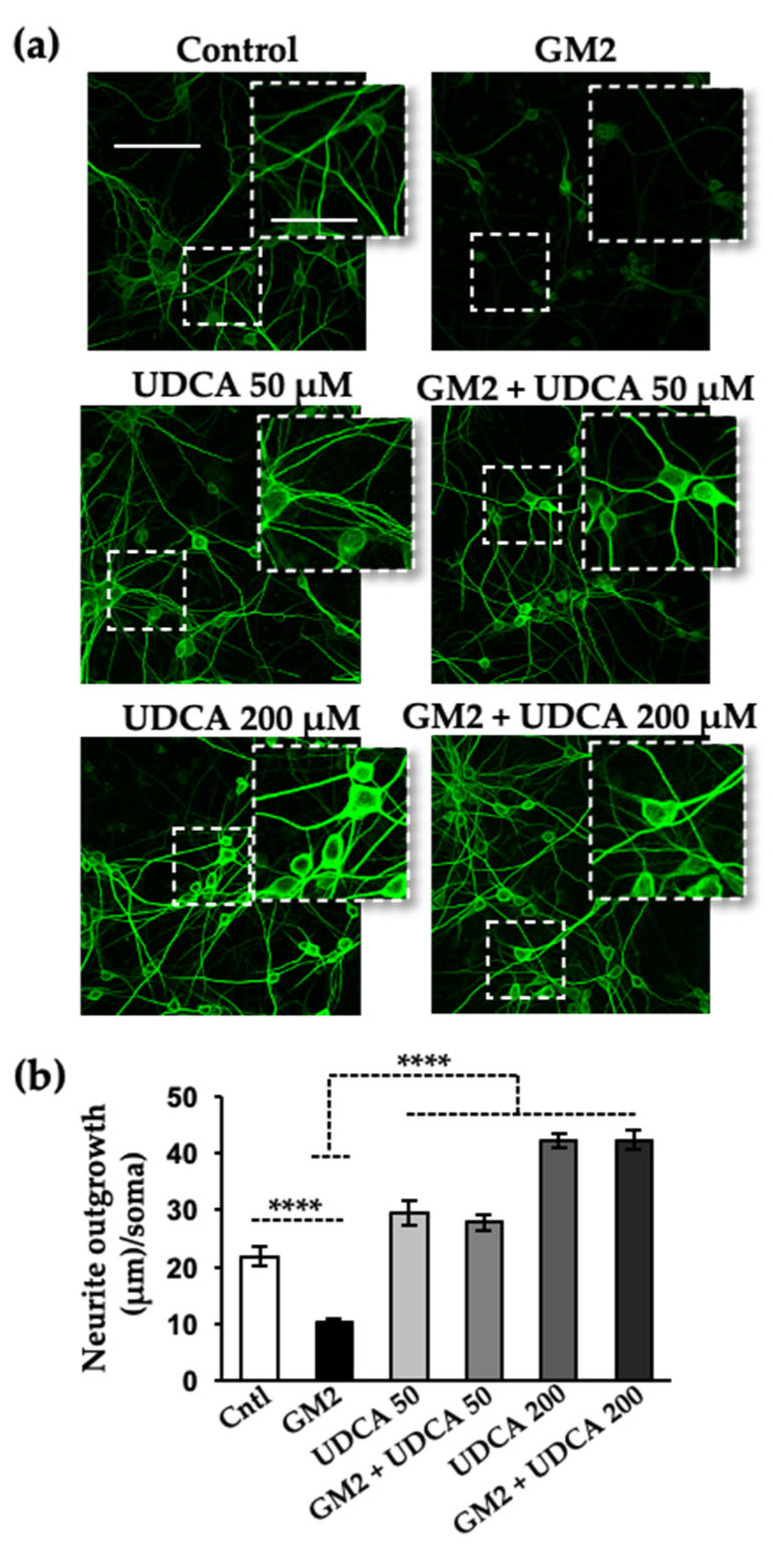
UDCA treatment reverses neuritic atrophy induced by GM2 accumulation. (**a**) Primary cortical neurons (10 DIV) were loaded with 2 μM of GM2 or left unloaded; after 1 h of incubation, the cells were treated with either vehicle, 50 μM or 200 μM of UDCA for an additional 19 h. At the end of this incubation, the cells were fixed with 4% paraformaldehyde, immunolabeled with mouse anti-MAP-2 antibody, and visualized with Alexa Fluor-488 conjugated secondary antibody. Images were recorded with a confocal microscope (Zeiss LSM 800). Scale bars: 100 μm (regular images), 50 μm (magnified images); (**b**) Histogram (mean ± SEM) represents neurite outgrowth with respect to total cells, analyzed with Image J plug-ins (NIH, USA). **** *p* ≤ 0.00001 with respect to the control as determined by one-way ANOVA, n = 3.

**Figure 2 ijms-24-07209-f002:**
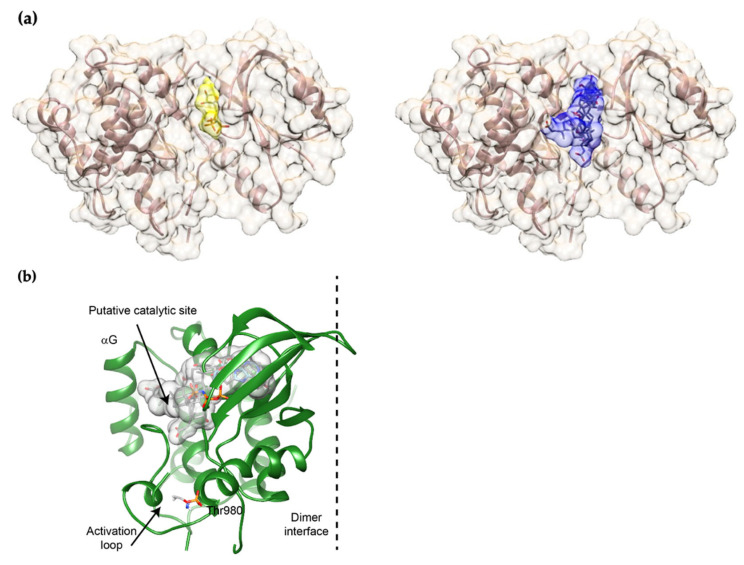
Docking models obtained on HADDOCK. (**a**) Left, model of mPERK bound to AMP-PNP, obtained on Chimera by combining the structures of PERK (3QD2) and PRK-AMP-PNP (2A19). For AMP-PNP, the surface appears yellow and sticks are colored by atom. Right, superposition of the surfaces of the four best docking models is shown in blue. Sticks colored by atom correspond to UDCA molecules; (**b**) Overall structure of the mPERK monomer. The threonine (Thr) 980 and the activation loop are shown. The α–G helix, which provides the docking site for eIF2α, is also depicted. Note that the binding site for UDCA matches the putative catalytic site.

**Figure 3 ijms-24-07209-f003:**
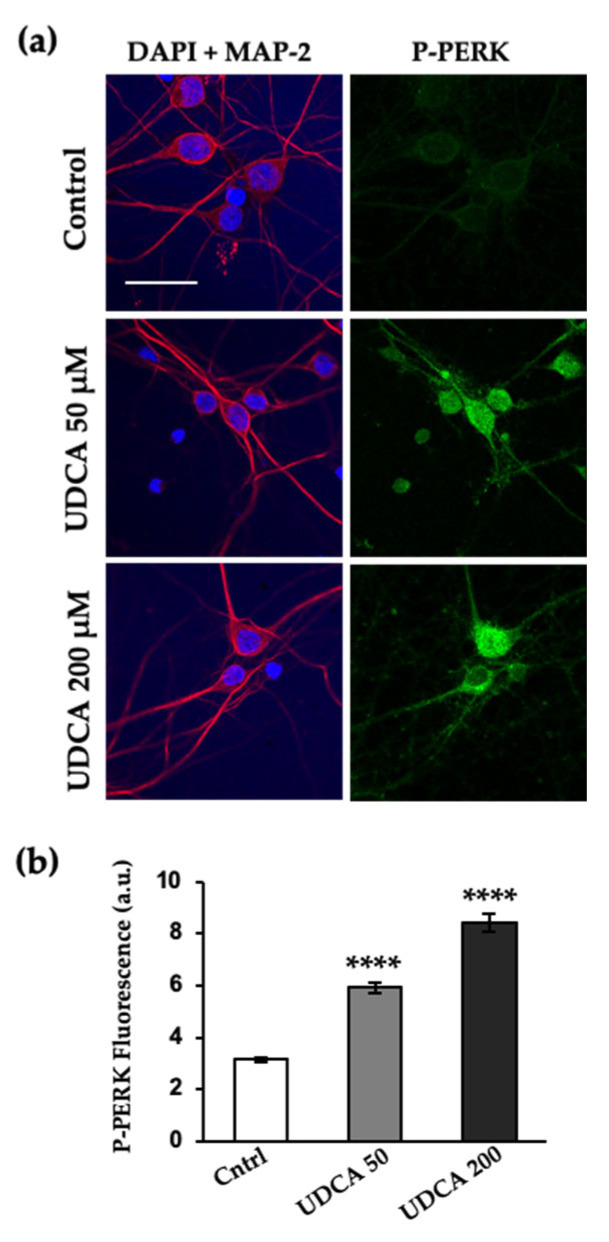
PERK phosphorylation levels are increased by UDCA treatment. (**a**) Primary cortical primary neurons (10 DIV) were incubated with or without 50 μM and 200 μM of UDCA for 19 h. The cells were fixed, immunolabeled with an anti-P-PERK antibody, and visualized with Alexa Fluor-488 conjugated secondary antibody. Nuclei were stained with DAPI (blue) and microtubule-associated protein 2 with chicken anti-MAP-2 antibody and visualized with Alexa-Fluor-594. Images were recorded with a confocal microscope. Scale bars: 30 μm; (**b**) Histogram (mean ± SEM) represents phospho-PERK fluorescence intensity, analyzed with Image J plug-ins (NIH, USA). **** *p* ≤ 0.00001 with respect to the control as determined by one-way ANOVA, n = 3.

**Figure 4 ijms-24-07209-f004:**
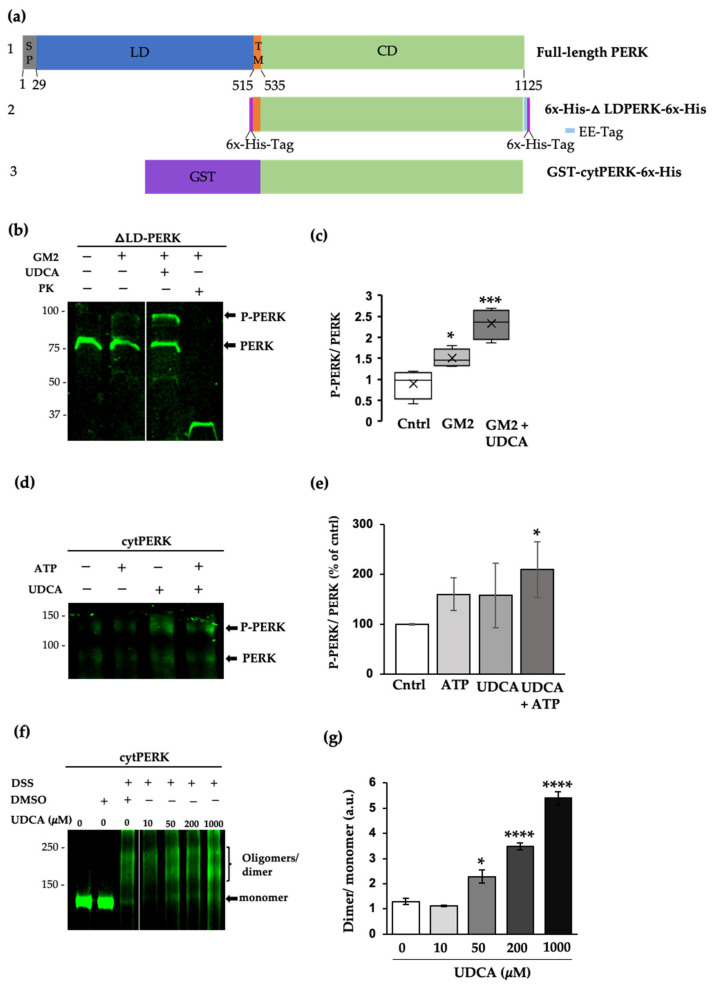
UDCA directly interacts with PERK and prompts its phosphorylation and oligomerization. (**a**) Diagrams: (1) Full-length mouse PERK provided as a reference; residue numbering based on UniProt Q9Z2B5. The signal peptide (SP) and the luminal (LD), transmembrane (TM), and cytosolic (CD) domains are indicated. (2) Hexahistidine-tagged PERK protein, lacking the luminal domain (ΔLD-PERK), was expressed in *E. coli* and used to constitute proteoliposomes. (3) The cytosolic domain of PERK (cytPERK) expressed in the bacteria as a GST fusion protein, and used in bulk solution reactions; (**b**) Immunodetection of PERK using an antibody that recognizes both the phosphorylated and non-phosphorylated forms. His-tagged ΔLD-PERK was incorporated into control or GM2 liposomes, purified by ultracentrifugation, and incubated with ATP (1 µM), in the presence or absence of UDCA (200 µM) for 20 min at 27 °C. Proteins were resolved on 10% SDS-PAGE; (**c**) Boxplots indicate median, 25th and 75th percentile limits, and extreme PERK phosphorylation values determined by the phosphorylated to non-phosphorylated ratio; (**d**) GST-cytPERK was incubated with ATP (0.1 mM), resolved on 7% SDS-PAGE, and probed with an anti-PERK antibody as in (**b**); (**e**) Histogram corresponding to densitometric analysis of P-PERK and PERK values normalized with control ratio value (100%); (**f**) Recombinant GST-cytPERK (1.3 µM) was incubated in the presence of DSS (0.4 mM) or DMSO (vehicle) and increasing concentrations of UDCA for 30 min at 27 °C. The reaction was quenched with 50 mM Tris-HCl (15 min at 27 °C). The proteins were resolved on 7% SDS-PAGE and probed as in (**b**); (**g**) Histogram corresponding to densitometric analysis of ratio values for dimer relative to monomer (**c**) * *p* ≤ 0.05, *** *p* ≤ 0.0001, n = 3; (**e**) * *p* ≤ 0.05, n = 3; (**g**) * *p* ≤ 0.05, **** *p* ≤ 0.00001, n = 4 (ANOVA, Tukey’s HSD test).

**Figure 5 ijms-24-07209-f005:**
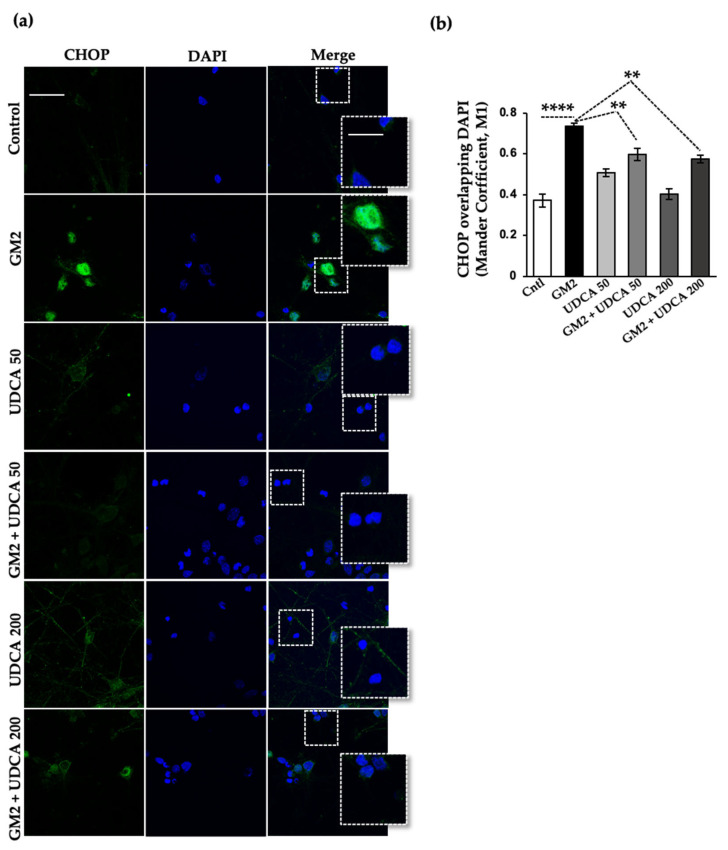
UDCA decreases the translocation of the transcription factor CHOP to the nuclei in GM2-stressed neurons. (**a**) Primary cortical neurons (10 DIV) were loaded with GM2, then treated with UDCA as in Figure 1. Fixed cells were labeled with anti-CHOP antibody (green) and their nuclei were labeled with DAPI (blue). Images were recorded with a confocal microscope. Scale bars: 30 μm (regular images), 15 μm (magnified images); (**b**) Histogram (mean ± SEM) represents Manders’ overlap coefficient, M1 (Channel 1: green, Channel 2: blue) calculated on Image J (NIH, USA). ** *p* ≤ 0.001; **** *p* ≤ 0.00001 with respect to the control as determined by one-way ANOVA, n = 3.

## Data Availability

The data that support the findings of this study are available from the corresponding author upon reasonable request.

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
