# Peer review of "Ursodeoxycholic Acid Binds PERK and Ameliorates Neurite Atrophy in a Cellular Model of GM2 Gangliosidosis"

_ijms, 2023, doi:10.3390/ijms24087209_

Round 1
Reviewer 1 Report
Morales and co-workers describe the potentiality of ursodeoxycholic acid to ameliorate the phenotype of GM2 gangliosidosis. The paper is well-written, they clearly present the results, and the topic interests the scientific community.
Please find some issues below.
Introduction
The ER stress players recently emerged as a potential therapeutic target in lysosomal storage diseases. Relevant papers on the topic should be cited:
- Doi: 10.1016/j.bbrc.2011.08.080
- Doi: 10.3390/ijms23095105
- Doi: 10.1093/hmg/ddm324
Results
2.1 Cells were treated for 20 hours with GM2, then with the vehicle or UDCA. Please, clarify how long the cells were treated with the drug.
2.3 Is the antibody anti-PERK or anti-P-PERK? Please, check the legend.
Figure 4b. Loading control is missing.
Methods
Apparently, all the experiments were performed using rat or mouse cell models, but this is not evident in the results and discussion. Please, clarify.
Supplementary materials are never cited in the text.
Reviewer 2 Report
Review of the paper "Ursodeoxycholic acid binds PERK and ameliorates neurite atrophy in a cellular model of GM2 gangliosidosis" presented by C. Morales, M. Fernandez, R. Ferrer, D. Raimunda, D C. Carrer and M. Bollo
The reviewed paper focuses on the search for an effective method of therapy of GM2 gangliosidosis, which characterizes neurodegenerative storage disorder such as Sandhoff , Tay-Sachs disease and many others. The molecular mechanisms through which GM2 buildup leads to neuronal death are still not fully understood. What is known is that PERK is a key factor to change neuron vulnerability to apoptosis associated with GM2 accumulation. Currently, there is no approved therapy for neuronal ganglioside storage diseases. In order to search for drugs capable of influencing neuronal death caused by GM2 accumulation, the authors of the reviewed paper have studied in detail the mechanism of interaction of PERC with ursodeoxycholic acid (UDCA), a bile acid that is able to modulate PERC activity in in vivo and in vitro models. The merit of the authors is the numerous studies both at the cellular level and in model systems, including docking simulations, that neurite atrophy induced by the abnormal accumulation of GM2 in primary neuron cultures might directly interact with the cytosolic domain of PERK, which promotes kinase auto-phosphorylation and stabilizes its dimerization.
The thoroughness of both cell culture of rat cortical neurons and liposome experiments performed using multiple methods is impressive, confirming conclusively that dimerization of full-length PERK is initiated by the stress-sensing luminal domain, and that perturbations in lipid membrane composition could further stimulate kinase activation. It is clearly shown that UDCA can directly interact with the cytosolic domain of PERK which engages the ATP pocket, stabilizing the active form of the kinase. The authors rightly argue that this mechanistic link may be exploited for the therapeutic purposes of GM2 gangliosidosis. The figures presented in the article are informative and the captions even include methodological details. However, the text of the section describing the results is somewhat overloaded with information about the authors' previous experiments. It would be desirable to include these fragments in the discussion in order to present the new results more clearly.
The report is well written, the introduction is clear, the aims and the hypothesis are correctly formulated. The peer-reviewed research is of a high standard, the results support the conclusions and the discussion is exhaustive.
After minor revisions the article may be published.
Round 2
Reviewer 1 Report
paper acceptable in the present form